# Transforming Vision Transformer: Towards Efficient Multi-Task Asynchronous Learning

**Hanwen Zhong**[1,2]   **Jiaxin Chen**[1,2*]   **Yutong Zhang**[1,2]   **Di Huang**[2]   **Yunhong Wang**[1,2]

[1]State Key Laboratory of Virtual Reality Technology and Systems, Beihang University, China
[2]School of Computer Science and Engineering, Beihang University, Beijing, China
{hanwenzhong,jiaxinchen,ytzhang_mq,dhuang,yhwang}@buaa.edu.cn

## Abstract

Multi-Task Learning (MTL) for Vision Transformer aims at enhancing the model capability by tackling multiple tasks simultaneously. Most recent works have predominantly focused on designing Mixture-of-Experts (MoE) structures and integrating Low-Rank Adaptation (LoRA) to efficiently perform multi-task learning. However, their rigid combination hampers both the optimization of MoE and the effectiveness of reparameterization of LoRA, leading to sub-optimal performance and low inference speed. In this work, we propose a novel approach dubbed Efficient Multi-Task Learning (EMTAL) by transforming a pre-trained Vision Transformer into an efficient multi-task learner during training, and reparameterizing the learned structure for efficient inference. Specifically, we firstly develop the MoEfied LoRA structure, which decomposes the pre-trained Transformer into a low-rank MoE structure and employ LoRA to fine-tune the parameters. Subsequently, we take into account the intrinsic asynchronous nature of multi-task learning and devise a learning Quality Retaining (QR) optimization mechanism, by leveraging the historical high-quality class logits to prevent a well-trained task from performance degradation. Finally, we design a router fading strategy to integrate the learned parameters into the original Transformer, archiving efficient inference. Extensive experiments on public benchmarks demonstrate the superiority of our method, compared to the state-of-the-art multi-task learning approaches. The project page is available at https://github.com/Yewen1486/EMTAL.

## 1 Introduction

Multi-task learning (MTL) [1, 2, 3] for Vision Transformer (ViT) aims at simultaneously learning multiple tasks, which has gained popularity in the computer vision community recently. It solves multiple relevant problems through sharing feature representations and forming a unified multi-task learner, thus enhancing the training and inference efficiency, and reducing the storage overhead. Moreover, by virtue of a well-designed MTL framework, the performance of each task can be further improved. Due to these merits, MTL has been used in a wide range of applications, such as the scene understanding [4, 5] and the erudite fine-grained recognition [6].

Despite both theoretical [7] and practical [8, 9] validations of their potential on enhancing the model generalizability, conventional MTL approaches [6, 10] often suffer performance degradation when compared to training tasks independently, which is primarily due to two reasons. Firstly, they adopt suboptimal learning strategies leading to conflicting task gradients and varying loss scales [11], which increase competitive interference during task optimization. Secondly, their model structures often fail to extract representative features for each task when utilizing a shared backbone network [12, 4, 6, 5].

---

*Corresponding author.

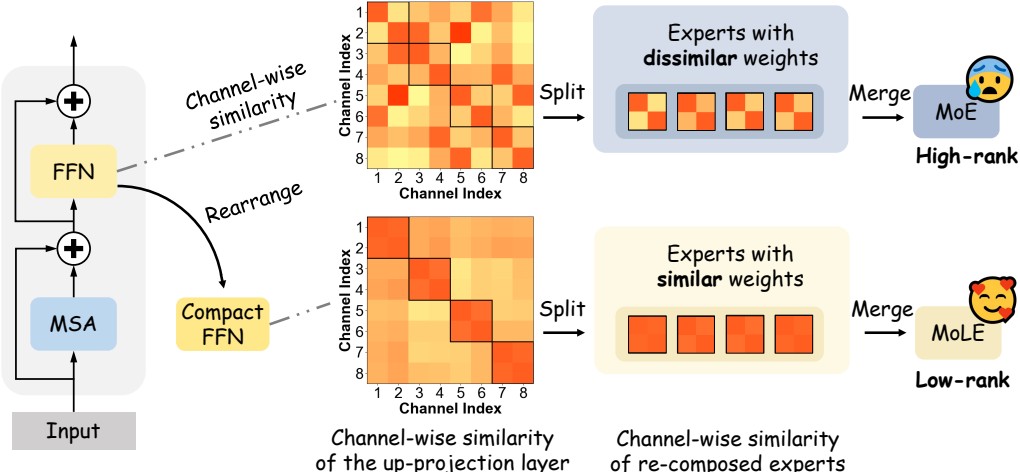

Figure 1: *FFN as Mixture of Low-rank Experts*. Given an up-projection weight matrix in FFN, a straightforward way of splitting it into MoE is to divide every $K$ channels into separate experts, resulting in highly dissimilar experts and a high-rank MoE, which is inherently unsuitable for integration with LoRA. In contrast, our proposed MoLE approach rearranges the weight matrix into groups of similar channels as experts, creating specialized low-rank experts that are better suited for integrating with LoRA.

Several recent works have attempted to deal with the above issues, which concentrate on the following two aspects. 1) ***Efficient multi-task learners.*** As shown in Figure 2, instead of designing complex but incompact network structures [12, 4] that incur a large number of tunable parameters, recent works [13, 5, 14] such as MLoRE and MOELoRA explore the advantages of Mixture-of-Experts (MoE) in extracting task-specific features by enhancing the diversity of parameters and features [15, 16, 17], and Parameter-Efficient Fine-Tuning (PEFT) in reducing the tunable parameters and storage overhead [18, 19, 20, 21]. Nevertheless, MLoRE [5] still relies on a substantial number of additional parameters, limiting the overall efficiency and feasibility of training. MOELoRA [14] adopts a unitary LoRA structure to tune the experts, which weakens the learning capability of individual experts. Moreover, both methods utilizes task-driven routers, requiring either a static network with a fixed number of tasks or a dynamic routing network. The former results in significant storage overhead, while the latter increases the inference cost. 2) ***Multi-Task Optimization (MTO) strategies***. Existing works on MTO can be broadly categorized into the gradient-based methods and the loss-based methods. The gradient-based methods [22, 2, 23, 3, 24, 25] seek to balance the gradients across multiple tasks in the last shared layer, by decreasing differences in their magnitude or direction, and aggregating sub-gradients into a unified one. The loss-based methods [26, 27, 28] optimize the MTL process by balancing the multi-task losses. Recently, IMTL [11] is proposed to treat all tasks equally without bias, while AMTL [29] synchronizes learning progress across tasks. However, as each task has its own intrinsic optimization pace due to varying levels of training difficulty for distinct tasks, forcing synchronization in MTO disrupts these inherent properties, thus leading to suboptimal solutions. For more detailed discussion of related works, we refer to Appendix A.

To overcome the drawbacks of existing works, we propose a novel MTL framework dubbed Efficient Multi-Task Asynchronous Learning (EMTAL). Basically, EMTAL consists of the MoEfied LoRA structure, the Quality Retaining (QR) optimization mechanism and the router fading strategy, of which MoEfied LoRA decomposes a pre-trained Vision Transformer model into an efficient multi-task learner, QR accomplishes asynchronous learning of multi-task knowledge and enable establishing an efficient unified model by combining with the router fading strategy. Specifically, inspired by MoEfication [30, 31], the proposed MoEfied LoRA firstly decomposes the FFN layer of Vision Transformer into a MoE structure by clustering similar channels into experts, creating specialized low-rank experts as demonstrated in Figure 1. Considering the inherent low-rank property of each expert, LoRA is naturally employed to perform efficient training of MoE. Subsequently, in order to

achieve asynchronous learning of multi-task knowledge based on the MoEfied LoRA module, QR constrains logits of early converged tasks to retain near the optima when continuously optimizing the insufficiently converged tasks, thus avoiding severe interference between tasks and improving multi-task optimization. Finally, the router fading strategy is combined with MoEfied LoRA and QR by gradually diminishing the router's fole in the last training epochs, seamlessly integrating learned parameters into the model structure without incurring extra inference time cost and storage overhead.

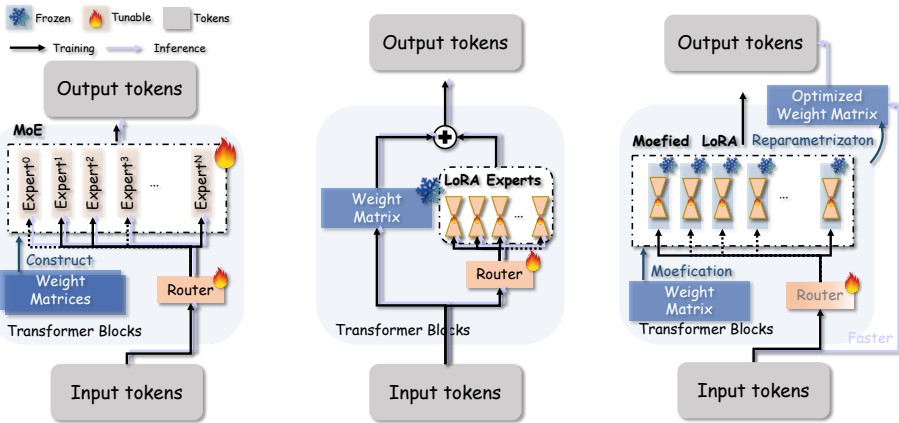

(a) MoE [12, 4]. The conventional MoE leverages multiple expert networks and a gating mechanism to dynamically select the most relevant experts for each input. It focuses on designing complex but incompact network structures, incurring a large number of tunable parameters.

(b) LoRA Experts [5, 14]. LoRA Experts employ unified low-rank adaptation modules to achieve the parameter efficiency, which however limit the expert capacity, and requires either static networks with substantial storage overhead or dynamic routers with high inference cost.

(c) MoEfied LoRA (Ours). Our method groups similar weights into specialized low-rank experts, enabling seamless integration with LoRA to create an efficient multi-task learner. Besides, by combing with a router fading strategy, our method ensures both training and inference efficiency while substantially reduces storage overhead.

Figure 2: Summary of representative architectures of multi-task learning.

The main contributions of this paper are summarized as follows. 1) We propose a novel efficient multi-task learning framework dubbed EMTAL. To the best of our knowledge, our work makes the first investigation on decomposing a pre-trained Vision Transformer model for multi-task learning and reparameterizing the learned multi-task knowledge into a unified model. 2) We design a MoEfied LoRA structure, a QR multi-task optimization mechanism combined with a router fading strategy to accomplish an efficient asynchronous multi-task learner. 3) We extensively evaluate the proposed method on challenging multi-task fine-grained visual classification datasets and the VTAB benchmark, and the experimental results demonstrate that our method significantly improves the performance of single-task learning and the state-of-the-art MTL approaches.

## 2 The Proposed Approach

In this section, we mainly introduce the preliminary concepts of Vision Transformer, and describe the technical details of the proposed EMTAL approach.

### 2.1 Preliminary

Given an input image $I \in \mathbb{R}^{3 \times H \times W}$, a standard Vision Transformer [32] model with $L$ layers first divides $I$ into $m$ non-overlapping patches, which are further fed into a patch embedding layer, generating $m$ $D$-dimensional visual tokens. After concatenating with a class token, the input tokens are finally formed as $X^0 \in \mathbb{R}^{(1+m) \times D}$. Each transformer layer contains a Multi-headed Self-Attention (MSA) [33] block, a Feed-Forward Networks (FFN) block and a Layer Normalziation (LN). The tokens of the $l$-th layer are generated based on those in the $(l-1)$-th layer formulated as below:

$$X^{l'} = \text{MSA}(\text{LN}(X^{l-1})) + X^{l-1}, \quad X^l = \text{FFN}(\text{LN}(X^{l'})) + X^{l'}. \tag{1}$$

Similar to [30, 31], our work mainly focuses on FFN, which usually consists of two linear layers $\{\boldsymbol{W}_{up} \in \mathbb{R}^{D \times (r \cdot D)}, \boldsymbol{b}_{up} \in \mathbb{R}^{r \cdot D}\}$, $\{\boldsymbol{W}_{down} \in \mathbb{R}^{(r \cdot D) \times D}, \boldsymbol{b}_{down} \in \mathbb{R}^D\}$ and a GELU activation operation, where $r$ represents the scaling factor. Accordingly, FNN processes the normalized input $\boldsymbol{X}_n^{l}{}' = \text{LN}(\boldsymbol{X}^{l'})$ as follows:

$$\boldsymbol{X}_{\text{FFN}}^l = \text{GELU}(\boldsymbol{X}_n^{l}{}'\boldsymbol{W}_{up} + \boldsymbol{b}_{up})\boldsymbol{W}_{down} + \boldsymbol{b}_{down}. \tag{2}$$

This procedure ensures the effective transformation and projection of the input through the encoder layers, enabling the prediction of the class probability distribution $y$ for downstream tasks.

## 2.2 Framework Overview

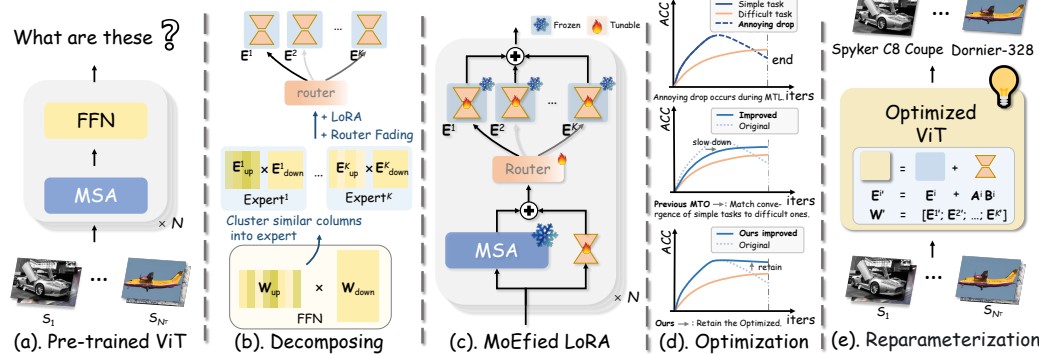

Figure 3: Illustration of the proposed EMTAL framework. Given a pre-trained ViT, we firstly decompose it into a MoE-based multi-task learner by using the balanced k-means. LoRA is then applied to the low-rank experts, creating an efficient multi-task learner dubbed MoEfied LoRA. During multi-task optimization, the Quality Retaining is employed to maintain the high-quality knowledge for tasks that have already converged. Finally, with the aid of the router fading strategy, the learned knowledge is reparameterized back into the pre-trained ViT, eliminating the extra inference cost.

As shown in Figure 3 (a), in order to establish a unified model for $N_T$ tasks, we follow [6] by unifying the label spaces for multiple tasks into an overall one with $N_{class}$ classes, and merging the training samples as $S = \bigcup_{t=1}^{N_T} S_t$, where $S_t$ denotes the set of training samples for the $i$-th task. Supposing a pre-trained Vision Transformer with an embedding backbone $\Phi(\cdot; \theta_\phi) : \mathcal{X} \to \mathcal{F}$, where $\theta_\phi$ represents the parameters to be frozen in the network, it maps an input $x \in \mathcal{X}$ to the feature space $\mathcal{F}$. We decompose it into an efficient multi-task learner dubbed **MoEfied-LoRA** denoted by $\Phi'(\cdot; [\theta_\phi; \theta_t])$, where $\theta_t$ indicates the set of newly employed tunable parameters and $[; ]$ refers to the concatenation operation. Subsequently, the **Quality Retaining** multi-task optimization mechanism as well as the router fading strategy are applied to asynchronously learning the tunable parameters $\theta_t$. Finally, we reparameterize $\theta_t$ into the original backbone, achieving an efficient unified model $\Phi(\cdot; \theta'_\phi)$.

## 2.3 MoEfied-LoRA

Basically, MoE-based learners benefits MTL in the following two ways. First, they enable dynamic encoding of different samples across tasks via a router and multiple experts, significantly enhancing feature diversity. Second, they reduce the number of parameters and computational cost, remarkably promoting the training and inference efficiency. However, early attempts include delicately designed MMoE [12], M³ViT [4] and Mod-Squad [13] fail to establish a unified structure and inevitably introduce additional inference overhead, considering that LoRA increases inference latency by 20-30% without reparameterization [34]. Recently, MoEfication methods [30, 31] aimed at "*group together the neurons that are often activated simultaneously*" have demonstrated promising performance in constructing effective MoE structures from pre-trained ViT models. Inspired by this, in this work we attempt to "*group together the neurons with similar weights*" to construct the MoE structure. As the corresponding experts naturally meet the low-rank conditions in LoRA [34], we can therefore obtain an excellent efficient multi-task learner by combining with LoRA. By employing a router fading strategy to preserve the reparameterization property of LoRA, we can further eliminate additional inference latency.

Specifically, as shown in Figure 3 (b), we fistly decompose the FFN into a MoLE (Mixture of Low-rank Experts) structure. To convert the FFN in the $l$-th layer into MoE, we draw on insights from [35, 36], which view the FFN as a memory bank that retains knowledge from pre-trained models. Each column of the up-projection matrix $\boldsymbol{W}_{up}$ serves as a key to be matched, which each row of the down-projection matrix $\boldsymbol{W}_{down}$ act as the corresponding value. Since similar keys in the $\boldsymbol{W}_{up}$ tend to serve similar functions, they should be grouped within the same expert. Formally, $\boldsymbol{W}_{up}$ is clustered column-wise into $K$ clusters by using the balanced $k$-means [37], with a mapping function $C(\cdot)$ that assigns the $idx$-th column to cluster $C(idx)$. Therefore, columns within the same cluster constitute an expert. Additionally, $\boldsymbol{b}_{up}, \boldsymbol{W}_{down}$ should match $\boldsymbol{W}_{up}$ channel by channel, and also adhere to the clustering results of $\boldsymbol{W}_{up}$ to construct the experts $\{\boldsymbol{E}^i\}_{i=1}^K$. Consequently, they are concatenated for the MoEfication process and then split to obtain the corresponding experts, formulated as below:

$$\boldsymbol{W} = [\boldsymbol{W}_{up}; \boldsymbol{b}_{up}; \boldsymbol{W}_{down}^{\mathsf{T}}], \boldsymbol{W} \in \mathbb{R}^{(2D+1) \times (rD)}, \tag{3}$$

$$\boldsymbol{E}^i = \boldsymbol{W}_{\{idx|C(idx)=i\}}, i \in 1, 2, \cdots, K, \boldsymbol{E}^i \in \mathbb{R}^{(2D+1) \times (\frac{rD}{K})}, \tag{4}$$

$$\boldsymbol{E}_{up}^i, \boldsymbol{E}_b^i, \boldsymbol{E}_{down}^i = \boldsymbol{E}_{1:D}^i, \boldsymbol{E}_D^i, \boldsymbol{E}_{D+1:2D+1}^i. \tag{5}$$

Since the column vectors within these experts are relatively similar, exhibiting low-rank characteristics, they naturally satisfy the low-rank conditions in LoRA [34]. Therefore, we apply a set of LoRA parameters $\{\boldsymbol{A}_{up}^i, \boldsymbol{B}_{up}^i\}$ and $\{\boldsymbol{A}_{down}^i, \boldsymbol{B}_{down}^i\}$ for $\boldsymbol{E}_{up}^i$ and $\boldsymbol{E}_{down}^i$ to enable more efficient learning and improved performance, which is referred as MoLE and formally described as below:

$$\boldsymbol{E}_{up}^i{}' = \boldsymbol{E}_{up}^i + \boldsymbol{A}_{up}^i \boldsymbol{B}_{up}^i, \tag{6}$$

$$\boldsymbol{E}_{down}^i{}' = \boldsymbol{E}_{down}^i + \boldsymbol{A}_{down}^i \boldsymbol{B}_{down}^i. \tag{7}$$

Moreover, to leverage the benefits of dynamic routing for extracting diverse features during training, we initially establish a sample-driven soft router for each MoLE, denote as $\boldsymbol{W}_r \in \mathbb{R}^{D \times K}$, to reweight the experts. For the $l$-th layer of MoLE, we calculate weights for each experts as $\omega^l$ by adopting the following formulation:

$$\omega^l = K \cdot \mathrm{softmax}\left(\frac{\mathrm{LN}(\boldsymbol{X}^{l'}) \boldsymbol{W}_r^l}{\tau}\right), \tag{8}$$

During training, the router is employed to fully optimize the MoLE. The output $\boldsymbol{X}_{\mathrm{FFN\text{-}de}}^l$ of the decomposed FFN is calculated as follows:

$$\boldsymbol{X}_{\mathrm{FFN\text{-}de}}^l = \sum_{i=1}^K \mathrm{GELU}([\omega_1^l \cdot (\boldsymbol{X}_n^{l}{}' \boldsymbol{E}_{up}^1{}' + \boldsymbol{E}_b^1); \cdots ; \omega_K^l \cdot (\boldsymbol{X}_n^{l}{}' \boldsymbol{E}_{up}^K{}' + \boldsymbol{E}_b^K)]) \boldsymbol{E}_{down}^i{}' + \boldsymbol{b}_{down}. \tag{9}$$

## 2.4 Quality Retaining Optimization

The goal of multi-task learning (MTL) is to ensure that the final model performs well across all tasks. However, due to distinct task difficulties and individual optimization schedules, achieving optimal performance on all tasks simultaneously with a single model can be challenging. As displayed in Figure 3 (d), despite introducing strong priors to synchronize the optimization schedules across tasks, existing methods [26, 29, 25] can disrupt the inherent schedules of tasks with varying difficulties, creating challenges for MTL optimization. Therefore, we propose a different perspective: maintaining the inherent optimization pace of each task is crucial. Specifically, we allow asynchronous convergence of tasks and introduce the Quality Retaining (QR) MTO strategy to preserve high-quality knowledge from already converged tasks during subsequent optimization.

Specifically, at iteration $iter$, we maintain an optimal knowledge bank $\boldsymbol{Z} \in \mathbb{R}^{N_{class} \times N_{class}}$, which records the Exponential Moving Average (EMA) logits of each class learned during the optimization process from iteration 0 to $iter - 1$. This knowledge is distilled into the currently optimized model using a distillation loss. Formally, we maintain the knowledge bank $\boldsymbol{Z}$ using EMA. For a sample $s$ in the current training batch with label $label_s$, we update $\boldsymbol{Z}_{label_s}^{iter}$ as follows:

$$\boldsymbol{Z}_{label_s}^{iter} = m \cdot \boldsymbol{Z}_{label_s}^{iter-1} + (1 - m) \cdot \boldsymbol{z}_s, \tag{10}$$

where $\boldsymbol{z}_s$ indicates the logits of sample $s$ and $m \in (0, 1)$ is a momentum coefficient. This results in a real-time updated knowledge repository $\boldsymbol{Z}$.

To ensure the retention of high-quality knowledge for already optimized tasks, we employ a straightforward method, *i.e.*, weighting the distillation process by the reciprocal of the loss from each task. This procedure implies that tasks with lower loss (already optimized) should rely more on the learned knowledge, while tasks with higher loss (still being optimized) will depend more on the ground truth. Therefore, the Quality Retaining loss $L_{QR}$ for samples within a mini-batch $S_{batch}$ is defined as below:

$$\mathcal{L}_{QR} = \sum_{t=1}^{N_T} \frac{1}{\mathcal{L}_{CE,t}} \cdot \sum_{s \in S_{batch}} \mathrm{KL}\left(\mathrm{softmax}(\boldsymbol{z}_s), \mathrm{softmax}(\boldsymbol{Z}_{label_s})\right) \cdot \mathbb{1}(s \in S_t), \qquad (11)$$

where $\mathrm{KL}(\cdot)$ and $\mathbb{1}$ indicates the Kullback-Leibler divergence [38] and the indicator function, respectively. $\mathcal{L}_{CE,t}$ is the Cross-Entropy loss for the $t$-th task.

Based on Eq. (11), the overall training loss is formulated as $\mathcal{L} = \sum_{t=1}^{N_T} \mathcal{L}_{CE,t} + \mathcal{L}_{QR}$. This strategy ensures that the model maintains optimal performance for already converged tasks while allowing other tasks to continue their optimization at their inherent pace.

## 2.5 Router Fading and Insights on the Unified Model Structure

MTL aims to develop a universal model capable of executing multiple tasks simultaneously. To achieve this goal, we transform the unified pre-trained model into an MoEfied LoRA and develop the quality retaining mechanism to preserve multi-task knowledge as discussed in the previous sections. However, the dynamic routing in MoEfied LoRA limits LoRA's capability of reparameterizing the learned parameters into a unified, static pre-trained structure. To address this issue, we design a router fading strategy that gradually diminishes the router's role in the later stages of training. This approach allows the knowledge embedded within the optimized router to be implicitly absorbed as follows:

$$\omega^l = \alpha * \omega^l + (1 - \alpha), \qquad (12)$$

where $\alpha \in [0, 1]$ is the trade-off hyper-parameter. Finally, as shown in Figure 3 (e), we completely remove the router after training by setting $\alpha = 0$. In the mean time, we reparametere knowledge learned by LoRA back using Eq. (6) and Eq. (7), and concatenate them to replace the original parameters $\boldsymbol{W}_{up}, \boldsymbol{b}_{up}, \boldsymbol{W}_{down}$ of the $l$-th layer with $\boldsymbol{W}_{up}', \boldsymbol{b}_{up}', \boldsymbol{W}_{down}'$ as below:

$$\boldsymbol{W}_{up}' = [\boldsymbol{E}_{up}^{1}{}'; \boldsymbol{E}_{up}^{2}{}'; \cdots; \boldsymbol{E}_{up}^{K}{}'], \qquad (13)$$

$$\boldsymbol{b}_{up}' = [\boldsymbol{E}_{b}^{1}{}'; \boldsymbol{E}_{b}^{2}{}'; \cdots; \boldsymbol{E}_{b}^{K}{}'], \qquad (14)$$

$$\boldsymbol{W}_{down}' = [\boldsymbol{E}_{down}^{1}{}'; \boldsymbol{E}_{down}^{2}{}'; \cdots; \boldsymbol{E}_{down}^{K}{}']. \qquad (15)$$

The above technique avoids extra computational costs and maintains a unified model structure, delivering a new perspective for multi-task learning.

# 3 Experimental Results and Analysis

## 3.1 Datasets and Evaluation Metric

By following [6], we mainly evaluate the performance of our proposed EMTAL method on the challenging *Multi-task FGVC* benchmark. In addition, we conduct experiments on the *Specialized VTAB-1k* dataset to validate the effectiveness over previous solutions. *Multi-task FGVC* is a collection of public datasets specifically for multi-task fine-grained visual classification, including CUB-200-2011 [39], Stanford Cars [40], FGVC-Aircraft [41] and Oxford Flowers [42]. In order to make fair comparisons, we adopt the standard training/testing split as depicted in [6]. *Specialized VTAB-1k* [43] consists of specialist images from specialized equipment, where we employ multi-task learning to fully leverage these expensively annotated data. We follow the standard training/validation splits used in [43] for fair comparisons. The top-1 accuracy is utilized as the evaluation metric. To further demonstrate the effectiveness of our method on the tasks of pixel-to-pixel dense prediction, we also conduct experiments on the NYUv2 dataset [44]. Specifically, we integrate our method

Table 1: Comparison of the top-1 accuracy (%) on the Multi-task FGVC benchmark, by using ViT-B/16 supervised pre-trained on ImageNet-21K. 'FT' denotes 'Full Fine-tuning'. The best results are highlighted in **bold** and the second best ones are underlined.

| Method | Reference | Unified Model | CUB-200 -2011 | Stanford Cars | FGVC- Aircraft | Oxford Flowers | Mean | Tunable Params. (M) | Inference Time (ms) |
|---|---|---|---|---|---|---|---|---|---|
| **MTL baselines** | | | | | | | | | |
| Separate FT | Baseline | ✗ | 86.4 | 87.6 | 77.2 | 98.8 | 87.49 | 343.92 | 14.30 |
| Union FT [6] | Baseline | ✓ | 83.1 | 90.7 | 78.3 | 97.5 | 87.39 | 85.98 | **7.15** |
| **MTO Gradient-based** | | | | | | | | | |
| Nash-MTL [24] | ICLR' 22 | ✓ | 88.3 | 90.2 | 80.6 | 99.5 | 89.65 | 2.82 | **7.15** |
| Aligned-MTL [25] | CVPR' 23 | ✓ | 88.9 | 90.6 | 81.6 | **99.7** | 90.17 | 2.82 | **7.15** |
| **MTO Loss-based** | | | | | | | | | |
| GLS [26] | CVPR' 19 | ✓ | 88.4 | 90.1 | 80.0 | 99.6 | 89.55 | 2.82 | **7.15** |
| AMTL [29] | ICCV' 23 | ✓ | 88.2 | 90.7 | 81.5 | **99.7** | 90.04 | 2.82 | **7.15** |
| **QR** | **Ours** | ✓ | 88.1 | **92.3** | 85.0 | 99.6 | 91.25 | 2.82 | **7.15** |
| **Efficient multi-task learners** | | | | | | | | | |
| Dual-Prompt [49] | ECCV' 22 | ✗ | 87.8 | 73.5 | 53.1 | 99.4 | 78.4 | 1.1 | 15.48 |
| Erudite [6] | CVPR' 23 | ✗ | 79.7 | 81.4 | 70.2 | 98.1 | 82.35 | 101.34 | 9.74 |
| MLoRE [5] | CVPR' 24 | ✗ | 74.8 | 59.5 | 49.9 | 99.2 | 70.76 | 188.01 | 42.02 |
| MOELoRA [14] | SIGIR' 24 | ✗ | 88.4 | 88.2 | 75,0 | **99.7** | 88.04 | 2.82 | 38.71 |
| **MoEfied LoRA** | **Ours** | ✓ | 88.5 | 91.3 | 81.5 | **99.7** | 90.27 | 1.20 | **7.15** |
| **EMTAL-1** | **Ours** | ✓ | **90.5** | 91.9 | 81.8 | **99.7** | 90.96 | **0.75** | **7.15** |
| **EMTAL-2** | **Ours** | ✓ | 90.0 | 92.2 | 83.5 | **99.7** | 91.35 | 0.90 | **7.15** |
| **EMTAL-4** | **Ours** | ✓ | 89.8 | **92.3** | **85.2** | **99.7** | **91.73** | 1.20 | **7.15** |

with TaskPrompter [45] by applying MoEfied LoRA and QR to the FFN layers and the semantic segmentation task head, respectively.

In addition, we evaluate on *Multi-task FGVC* for few-shot learning under 1, 2, 4, 8, and 16 shots, by following existing works [46, 47].

### 3.2 Implementation Details

We utilize ViT-B/16 [2] pre-trained on ImageNet-21K [32] as the base model. We use the AdamW optimizer [48] to fine-tune our models for 100 epochs and adopt the cosine learning rate decay with a linear warm-up for 10 epochs in all experiments. We fix the hyper-parameters $\tau$ in Eq. (8) to 5, since it exhibits stable performance with distinct values. As for data augmentation, we employ random resize cropping to $224 \times 224$ pixels and a random horizontal flip during training and resize to $248 \times 248$ pixels with a center crop to $224 \times 224$ pixels. All experiments are conducted on a single Nvidia GeForce RTX 3090 GPU.

### 3.3 Comparison with the State-of-the-Art Approaches

To comprehensively evaluate the performance of our approach, we compare with the MTL full fine-tuning baseline and the following categories of state-of-the-art approaches: 1) MTO Loss-based methods, including GLS [26] and AMTL [29]; 2) MTO Gradient-based methods, including Nash-MTL [24] and Aligned-MTL [25] combined with vanilla LoRA-16; 3) Efficient multi-task learners methods, including Dual-Prompt [49], Erudite [6], MLoRE [5] and MOELoRA [14]. As the performance of EMTAL depends on the inherent low-rank properties, we report the results of our method using distinct ranks including 1, 2 and 4, denoted by EMTAL-1, EMTAL-2 and EMTAL-4, respectively.

As summarized in Table 1, the proposed EMTAL method consistently improves the performance at different ranks, promoting the top-1 accuracy of the Separate full-finetuning baseline by an average of 3.47%, 3.86% and 4.24%, respectively. More importantly, EMTAL tunes only a negligible amount of parameters (*i.e.*, 1.20M) compared to the original model (*i.e.*, 343.92M), and incurs no extra inference

---
[2]ViT-B/16 supervised pre-trained on ImageNet-21K.

Table 2: Comparison results (%) with the state-of-the-art PEFT and MTL methods on Specialized VTAB-1k by using ViT-B/16 models supervised pre-trained on ImageNet-21K. 'FT' denotes 'Full Fine-tuning'. The best results are highlighted in **bold** and the second best is underlined.

| Method | Reference | Patch Camelyon | EuroSAT | Resisc45 | Retinopathy | Mean | Tunable Params. (M) |
|---|---|---|---|---|---|---|---|
| **MTL baselines** | | | | | | | |
| Separate FT | Baseline | 79.7 | 95.7 | 84.2 | 73.9 | 83.38 | 343.36 |
| Union FT [6] | Baseline | 84.3 | 93.9 | 83.0 | 75.2 | 84.08 | 85.99 |
| **Traditional PEFT** | | | | | | | |
| Adapter [50] | ICML' 19 | 76.3 | 88.0 | 73.1 | 70.5 | 76.98 | 1.08 |
| LoRA [34] | ICLR' 22 | 85.5 | 95.3 | 86.1 | 75.3 | 85.50 | 2.41 |
| VPT-D [18] | ECCV' 22 | 81.8 | **96.1** | 83.4 | 68.4 | 82.42 | 2.40 |
| SSF [19] | NeurIPS' 22 | **87.4** | 95.9 | 87.4 | 75.5 | 86.56 | 0.96 |
| SPT-L [51] | ICCV' 23 | 85.7 | **96.2** | 85.9 | 75.9 | 85.92 | 2.16 |
| ARC [20] | NeurIPS' 23 | 84.9 | 95.7 | 86.7 | 75.8 | 85.78 | 0.52 |
| **PEFT for MTL** | | | | | | | |
| AMTL [25] | ICCV' 23 | 86.4 | 95.0 | 85.8 | 75.9 | 85.79 | 2.41 |
| MOELoRA [14] | SIGIR' 24 | 86.9 | 96.1 | 88.4 | 76.7 | 87.01 | 2.41 |
| **EMTAL-1** | **Ours** | 85.5 | **96.2** | 88.0 | 77.5 | 86.78 | **0.34** |
| **EMTAL-2** | **Ours** | 87.3 | 95.7 | 88.1 | 78.7 | 87.43 | 0.49 |
| **EMTAL-4** | **Ours** | **87.4** | 96.1 | **89.1** | **78.9** | **87.89** | 0.78 |

Table 3: More evaluation results on NYUv2 with ViT-B/16. The best results are highlighted in **bold**.

| Method | Semseg mIoU ↑ | Depth RMSE ↓ | Normal mErr ↓ | Boundary odsF ↑ | Mean Δ (%) ↑ |
|---|---|---|---|---|---|
| TaskPrompter-Base [45] | 50.40 | 0.5402 | **18.91** | **77.60** | - |
| **+ EMTAL (Ours)** | **52.90** | **0.5284** | 18.95 | 77.10 | **1.57** |

cost, implying that our framework is significantly more effective and efficient than the traditional separate training paradigm.

Moreover, EMTAL consists of a reparameterizable and efficient multi-task learner, a Quality Retaining MTO mechanism and a router fading strategy. Compared to the state-of-the-art methods, each component of our approach shows significant advantages. In terms of the design of MTL structures, MoEfied LoRA seamlessly integrates low-rank experts with LoRA, resulting in improved performance. Furthermore, the router fading strategy and the reparameterization effectively reduce inference time, significantly enhancing the overall efficiency. Meanwhile, by considering the intrinsic task-specific optimization pace, the QR mechanism clearly improves previous MTO strategies. Overall, it utilizes a sample-driven router and multiple experts to extract diverse feature representations while preserving high-quality knowledge during training, making it highly beneficial for multi-task learning. In this way, our method achieves the highest accuracy compared to the state-of-the-art approaches, surpassing the second best Aligned-MTL by 1.56% while tuning fewer parameters.

We further evaluate the performance on *Specialized VTAB-1k*, where MTL is highly beneficial against previous solutions. As displayed in Table 2, some existing works focus on applying parameter-efficient fine-tuning on each task individually to avoid over-fitting. However, we find that performance can be significantly enhanced by incorporating multi-task learning. Specifically, applying AMTL and MOELoRA on the vanilla LoRA yields improvements of 0.29% and 1.51%, respectively. Furthermore, when directly applying our EMTAL to these tasks, we observe consistent improvements across different ranks, achieving the highest accuracy compared to the state-of-the-art approaches. Notably, it enhances the overall performance by 1.43%, while utilizing fewer parameters. In addition, on the NYUv2 dataset, our method significantly enhances performance in semantic segmentation and depth estimation tasks, while achieving comparable results in surface normal estimation and object boundary detection tasks. Overall, this led to an average relative improvement of 1.57%, validating the effectiveness of our approach for pixel-level prediction tasks. We provide more results, including using self-supervised pre-trained model DINOv2-large and the few-shot learning in Appendix B and Appendix C, respectively.

## 3.4 Ablation Study

In this section, we evaluate the effectiveness of the proposed main components, *i.e.* MoEfied LoRA and Quality Retaining by extensive ablation studies.

Table 4: Ablation results (%) of the main components on the Multi-task FGVC by using ViT-B/16 backbone. The best results are highlighted in **bold**.

| MoEfied LoRA | Quality Retaining | CUB-200 -2011 | Stanford Cars | FGVC- Aircraft | Oxford Flowers | Mean | Tunable Params. (M) |
|---|---|---|---|---|---|---|---|
| ✗ | ✗ | 83.1 | 90.7 | 78.3 | 97.5 | 87.39 | 86.26 |
| ✓ | ✗ | 88.5 | 91.3 | 81.5 | **99.7** | 90.27 | **1.20** |
| ✗ | ✓ | 88.5 | 91.6 | 84.4 | 99.6 | 91.04 | 86.26 |
| ✓ | ✓ | **89.8** | **92.3** | **85.2** | **99.7** | **91.73** | **1.20** |

Table 5: Ablation results (%) of the MoEfied LoRA and router fading strategy on Multi-task FGVC by using ViT-B/16 backbone. The best results are highlighted in **bold**.

| Method | CUB-200 -2011 | Stanford Cars | FGVC- Aircraft | Oxford Flowers | Mean | Tunable Params. (M) | Inference Time (ms) |
|---|---|---|---|---|---|---|---|
| Union FT | 83.1 | 90.7 | 78.3 | 97.5 | 87.39 | 86.26 | **7.15** |
| +MoEfied | 85.2 | 91.3 | 80.8 | 98.7 | 89.20 | 86.40 | 13.87 |
| +LoRA [34] | 88.2 | 91.0 | **81.7** | 99.6 | 90.14 | **1.20** | 13.87 |
| +Router fading | **88.5** | **91.3** | 81.5 | **99.7** | **90.27** | **1.20** | **7.15** |

Table 6: Ablation results (%) of the proposed MoEfied LoRA with different numbers of clusters (*i.e.* $k$) and distinct ways to construct experts.

| Hyper. | Clusters # | | | | | Method | | |
|---|---|---|---|---|---|---|---|---|
| | 1 | 4 | 16 | 64 | 192 | Co-activation | Gradient-cluster | **Ours** |
| Params. (M) | 1.05 | 1.09 | 1.20 | 1.64 | 2.82 | 1.20 | 1.20 | 1.20 |
| Mean Acc | 88.83 | 89.12 | **90.27** | 89.34 | 89.02 | 89.30 | 89.34 | **90.27** |

**Effect of the Main Components.** We evaluate the proposed components across the *Multi-task FGVC* Benchmark based on ViT-B/16. As Table 4 shows, MoEfied LoRA consistently boost the performance, achieving an average of 2.88% improvement with less tunable parameters. The results indicate that MoEfied LoRA is significantly beneficial to multi-task learning by providing more diverse features. In the mean time, the Quality Retaining MTO mechanism can further remarkably promote the accuracy, with a 3.65% improvement on average. A combination of these two components, *i.e.* MoEfied LoRA and QR, further boosts the overall performance across datasets, implying that MoEfied LoRA and Quality Retaining are complementary in multi-task learning.

**On MoEfied LoRA and Router Fading.** We further validate the effectiveness of MoEfied LoRA and the router fading strategy across the *Multi-task FGVC* benchmark based on ViT-B/16. Initially, we begin by decomposing the pre-trained model into a MOLE structure. A straightforward clustering and splitting of the FFN, combined with a sample-driven router, can achieve a 1.81% improvement by enhancing the the diversity of feature representations. Furthermore, applying LoRA to the low-rank experts yields a 0.94% gain in performance, as the small amount of tunable parameters reduces the risk of overfitting, and the the low-rank property of the experts aligns well with LoRA. Additionally, the proposed router fading strategy gradually diminishes the influence of the router over 50 epochs, effectively preserving the reparameterizable nature of LoRA and reducing the inference time.

Moreover, we conduct more ablation studies on the proposed MoEfied LoRA. The number of clusters $k$ significantly influence the performance of MoEfied LoRA, considering that a large number of clusters intends to incur simple experts with few channels, and a small number of clusters results in high-rank experts, either of which degrades the effectiveness of MoEfied LoRA. We empirically study the effect of $k$ by using 1, 4, 6, 64 and 192 clusters. As Table 6 displays, MoEfied LoRA reaches the highest accuracy when $k = 16$. Moreover, we compare different ways to construct experts, including the co-activation clustering [28] that groups weights based on activations for each channel and the gradient-cluster [47] that clusters weights according to cumulative gradients. As shown in Table 6, our method achieves the best performance, clearly demonstrating its effectiveness.

## 3.5 Visualization on the Low-rank Property of MoLE

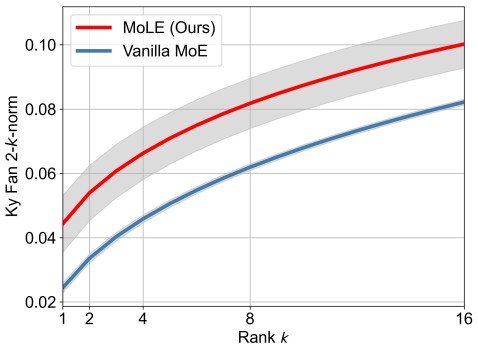 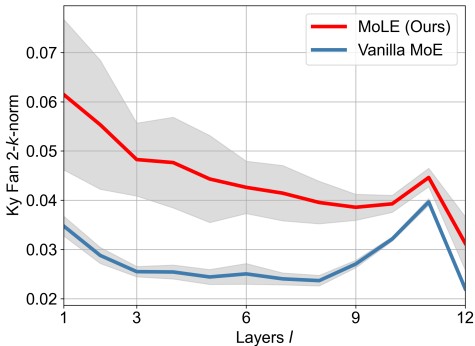

(a) Low-rank properties of experts across different ranks in the 4-th transformer block.

(b) Low-rank properties of experts across different transformer block $l$ with a fixed rank 1.

Figure 4: Comparison of the low-rank properties by using the vanilla MoE and the proposed MoLE, based on the Ky Fan 2-k norm [52]. A higher value signifies a stronger low-rank property.

As shown in Figure 4 (a), we measure the low-rank properties of experts by using the Ky Fan 2-k norm [52], and the results indicate that the experts generated by our method consistently exhibit statistically more significant low-rank properties across distinct ranks. Additionally, we analyze the low-rank properties of experts across different layers of ViT when the ranks is fixed as 1. As Figure 4 (b) demonstrates, the low-rank properties of experts are more significant in lower layers that those in higher layers.

We kindly refer to Appendix E for more detailed discussion about the broader impacts and limitations of our work .

## 4 Conclusion

In this paper, we focus on decomposing a pre-trained Vision Transformer model for multi-task learning, reparameterizing the learned multi-task knowledge back into the original model and establishing a unified model. We propose a novel efficient multi-task learning framework dubbed EMTAL, which mainly consists of the MoEfied LoRA module, the Quality Retaining (QR) mechanism and the router fading strategy. Concretely, MoEfied LoRA decomposes a pre-trained ViT into multi-task learners by clustering similar weight of FFN into experts and applies LoRA to tune the experts with low-rank properties. Subsequently, we leverage the inherent asynchronous convergence property of tasks and employ QR to preserve the optimized performance for converged tasks. Finally, the router fading strategy is introduced to eliminate extra inference cost. Extensive experiments on the public benchmarks demonstrate that our method substantially promotes performance by comparing with the state-of-the-art multi-task learning approaches, while also being effective in few-shot learning with limited data. Our work delivers a new perspective for efficient multi-task learning by decomposing a pre-trained model and reparameterizing back with a low rank updating.

## Acknowledgments

This work was partly supported by the National Natural Science Foundation of China (Nos. 62202034, 62176012, 62022011), the Beijing Natural Science Foundation (No. 4242044), the Beijing Municipal Science and Technology Project (No. Z231100010323002), the Aeronautical Science Foundation of China (2023Z071051002), the Research Program of State Key Laboratory of Virtual Reality Technology and Systems, and the Fundamental Research Funds for the Central Universities.

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

# A  Related Work

## A.1  Multi-Task Learning

Multi-Task Learning (MTL) [53] aims to improve the generalization performance of models across individual tasks by leveraging shared representations to exploit the commonalities and interdependencies between tasks. This approach also reduces the number of parameters and accelerates both training and inference [53, 54, 55]. Existing works are roughly divided into two categories: 1) the efficient multi-task learners and the multi-task optimization methods [56].

**Efficient multi-task learners.** These approaches concentrate on analyzing the impact of parameter sharing within models and can be broadly categorized into encoder-focused and decoder-focused approaches, depending on where information is exchanged or shared between tasks [56]. In encoder-focused models, task parameters are shared exclusively within the encoder to extract general features, by leveraging mechanisms such as feature fusion [57, 58, 59], attention [28, 6], and dynamic branching [60, 61], while the decoder consists of independent task-specific heads with no cross-task information sharing [22, 62, 1]. In decoder-focused models, parameters are shared across tasks within the decoder. The model initially makes separate predictions for each task and then refines these results by leveraging inter-task correlations through mechanisms such as multi-model distillation [63, 64, 65, 66], sequential task prediction [67], or cross-task consistency [68]. Some continual learning frameworks [69, 49] are also applicable to MTL, incorporating task-specific and task-shared parameters to achieve effective isolation and sharing between tasks.

**Multi-Task Optimization Methods.** The optimization-based methods [22, 70, 71, 62, 72, 25, 24] focus on balancing how tasks are learned, exploring effective solutions from the perspective of model optimization. These approaches enhance the optimization process of MTL through various design of multi-task losses [26, 68, 62, 73, 74, 29], which assigns appropriate loss weights to minimize conflicts among tasks. [75] proposes recording the optimal checkpoint for each task and learning from it by distilling the soft labels of individual samples. However, this approach may record a local optimal solution, leading to suboptimal results. Gradient manipulations [70, 22, 23, 2, 3] techniques address task interference by directly adjusting gradients, with recent methods emphasizing the formulation of a unified gradient vector subject to diverse constraints.

## A.2  Mixture-of-Experts

Mixture-of-Experts (MoE) [16, 15] are originally designed to combine the decisions of a series of sub-models, *i.e.*, expert networks, on the same input, enhancing conditional computational capabilities and enabling the scaling of parameters in neural networks [76, 77, 17, 78]. Therefore, MoE trains multiple specialized expert networks, each of which maintains its own unique set of trainable parameters. This design allows the expert networks to develop distinct internal representations tailored to their respective input data. Additionally, MoE employs a router that dynamically weights the outputs of each expert network, enabling their contributions to be combined into the final output.

MoE is also applied in Multi-Task Learning (MTL), effectively partitioning the parameter space and leveraging relevant model components for different tasks, making it a promising solution for MTL [15, 16, 17]. $M^3ViT$ [4] customizes the MoE layer within a ViT backbone, and activates task-specific experts during training to mitigate gradient conflicts in MTL. Mod-Squad [13] introduces a modular multi-task leaner based on MoE, along with a novel loss function, to address the gradient conflicts among tasks. MMoE [12] designs a multi-gate MoE to ensemble expert networks for various census analysis tasks, each with a different router. TaskExpert [79] generates multi-task predictions for all tasks in a single forward pass simultaneously, leading to significantly higher multi-task training efficiency. MLoRE [5] explicitly builds global relationships among all tasks within the MoE structure and introduces low-rank experts, improving the efficiency of MoE compared to the vanilla approach.

## A.3  Low-Rank Updating

Low-Rank Adaption (LoRA) [34] is a parameter-efficient fine-tuning method [34, 80, 81, 82], inspired by the observation from [83] that the difference in weights between the pre-trained model and the adapted model lies in a low intrinsic rank. With the success of low-rank structure [84, 85, 86, 87] as a parameter-efficient fine-tuning technique, numerous studies have demonstrated impressive results by combining LoRA and MoE for more efficient and effective model tuning. LoRAHub [88] first

trains multiple LoRA weight modules on upstream tasks. To adapt the model to a downstream task, it employs a gradient-free optimization method to determine the optimal coefficients for linearly combining the pre-trained LoRA modules. MOELoRA [14] utilizes a router network conditioned on a task identifier to dynamically combine the outputs of multiple LoRA experts. Similarly, MoCLE [89] designs a router network that is conditioned on the clustering information extracted from each input sample. LoRAMoE [90] splits the LoRA experts into two groups and explicitly learns distinct capabilities for each group. While these mixture-of-LoRA methods densely combine multiple LoRA experts, a sparse mixture of LoRA experts offers a more economical alternative, achieving comparable performance while maintaining roughly constant training and inference costs. The Octavius [91] method, for instance, selects the top-2 LoRA experts based on a router that conditions on the entire input instance, representing a more coarse-grained routing mechanism. Besides, earlier MTL methods use the low-rank structure to model task-generic features and generate task-specific features through linear combinations. VL-Adapter[82] exploits adapter-based methods to efficiently fine-tune generative models in a multi-task setting, while [87] introduces a framework for training multiple neural networks simultaneously, sharing all shareable layers and learning the sharing strategy in a data-driven manner.

# B More Experimental Results on DINOv2-large

Table 7: Comparison results (%) with the state-of-the-art PEFT approaches on the Specialized VTAB-1k benchmark by using the self-supervised pre-training ViT-L/14 model, *i.e.*, DINOv2-large [92]. 'FT' denotes 'Full Fine-tuning'. The best results are highlighted in **bold** and the second best ones are underlined.

| Method | Reference | Unified Model | Patch Camelyon | EuroSAT | Resisc45 | Retinopathy | Mean | Tunable Params. (M) |
|---|---|---|---|---|---|---|---|---|
| Separate FT | Baseline | ✗ | 88.1 | 96.1 | 90.9 | 77.2 | 88.08 | 1217.6 |
| BitFit [93] | ACL' 22 | ✗ | 85.2 | 96.1 | 90.7 | 75.7 | 86.92 | 1.08 |
| FacTtt [80] | AAAI' 23 | ✗ | 87.1 | 94.3 | 88.7 | 74.0 | 86.02 | **0.48** |
| FacTtk [80] | AAAI' 23 | ✗ | 86.1 | 94.6 | 89.5 | 74.2 | 86.10 | **0.48** |
| LoRA [34] | ICLR' 22 | ✗ | 88.3 | 96.4 | 91.4 | 77.4 | 88.38 | 7.08 |
| GLoRA [94] | arXiv' 23 | ✗ | 85.9 | 96.0 | 91.0 | 76.2 | 87.27 | 19.48 |
| OFT [95] | NeurIPS' 23 | ✗ | 88.4 | 96.4 | 91.5 | 77.2 | 88.38 | 8.40 |
| BOFT [96] | ICLR' 24 | ✗ | 88.9 | **96.6** | 91.6 | 77.3 | 88.60 | 7.96 |
| **EMTAL-4** | **Ours** | ✓ | **89.4** | 95.9 | **91.7** | **80.1** | **89.27** | 2.03 |

In addition to the ViT-B/16 pre-trained model as displayed in Table 2, we evaluate the performance of our method based on the self-supervised pre-training ViT-L/14 model, *i.e.*, DINOv2-large [3]. As summarized in Table 7, our method consistently outperforms the compared approaches. Notably, compared with the separate full-finetuning (Separate FT) baseline which incurs 1217.6M tunable parameters ($4 \times 304.4$M) and the other alternative methods, EMTAL successfully constructs a unified multi-task framework on this benchmark. Our approach eliminates the need to develop multiple specialized models while delivering superior performance, thereby validating its generalizability when applied to larger self-supervised models.

# C More Experimental Results on Few-shot learning

We also evaluate the performance of our method on the challenging few-shot learning task. As illustrated in Figure 5, the following observations can be made regarding the average performance : (i) EMTAL, LoRA, Adapter, and VPT exhibit similar performance with extreme limited amount of training data, such as with only 1 or 2 shots. (ii) As the amount of training data increases, EMTAL demonstrates a significant advantage. For instance, with 16 shots, EMTAL promotes the accuracy of the second best one by approximately 5%. This highlights that multi-task training has a particularly significant advantage over the separate training, especially in scenarios where reliable annotation information is limited.

---

[3]DINOv2-large.

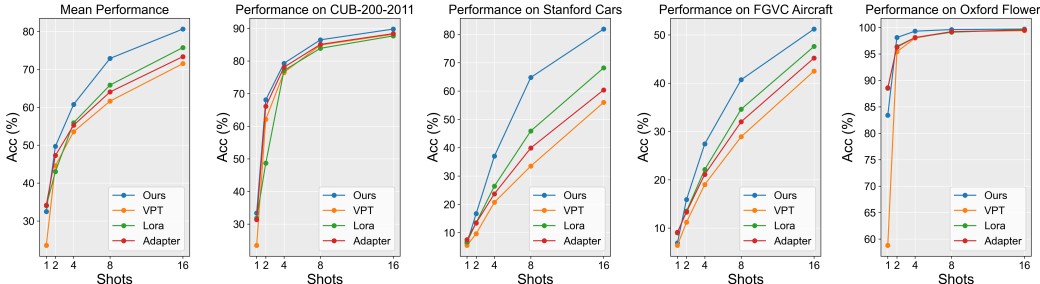

Figure 5: Comparison results using various separate training approaches in the context of few-shot learning on the multi-task FGVC datasets.

# D    Detailed Descriptions for the Evaluation Datasets

We provide detailed descriptions about the datasets used for evaluation. The train/val/test splits and the number of classes are summarized in Table 8.

Table 8: The statistics of the datasets used for evaluation. The train/val/test splits are the same as depicted in [6]. '-' indicates that the corresponding split is not available.

| Dataset | Description | #Classes | Training set | Val. set | Test set |
|---|---|---|---|---|---|
| Multi-task FGVC | | | | | |
| CUB-200-2011 [39] | Bird species recognition | 200 | 5,994 | - | 5,794 |
| Stanford Cars [40] | Car classification | 196 | 8,144 | - | 8,041 |
| FGVC-Aircraft [41] | Aircraft classification | 100 | 6,667 | - | 3,333 |
| Oxford Flowers [42] | Flower species recognition | 102 | 2,040 | - | 6,149 |
| Specialized VTAB-1k [43] | | | | | |
| Patch Camelyon [97] | | 2 | | | 32,768 |
| EuroSAT [98] | Specialized | 10 | 800/1,000 | 200 | 5,400 |
| Resisc45 [99] | | 45 | | | 6,300 |
| Retinopathy [100] | | 5 | | | 42,670 |

By following Erudite [6], we employ the Multi-task Fine-Grained Visual Classification datasets to evaluate the performance of our proposed EMTAL, which consists of CUB-200-2011 [39], Stanford Cars [40], FGVC-Aircraft [41] and Oxford Flowers [42]. We also employ *Specialized VTAB-1k* [43], consisting of specialist images from specialized equipment, where multi-task learning is applied to fully leverage these annotated data.

# E    Limitations and Broader Impacts

Our work may have the following potential impacts. Firstly, compared to traditional multi-task learning approaches, EMTAL maximizes the use of limited data for downstream multi-task learning. This capability facilitates the rapid transfer of large models pre-trained on vast datasets to downstream tasks, significantly conserving computational resources. Secondly, our method is designed based on reparameterization, allowing the model to be transferred to downstream tasks without altering the deployed backbone architecture. This approach, which involves simply replacing a set of weights, is more convenient than many multi-task learning methods [4, 13, 5].

Regarding limitations, our model structure currently employs a unified rank across all experts. Despite that this design benefits parallel computing during training, our experiments reveal that different tasks have varying difficulties, necessitating different optimal ranks. For example, the optimal rank for the CUB-200-2011 dataset is one, whereas for the Stanford Cars dataset, it is four. Thus, dynamically and adaptively selecting the appropriate ranks for different tasks to improve multi-task learning is a promising research direction. Additionally, we observed that the low-rank property of shallow experts is more pronounced than that of deep experts in the network, suggesting that dynamically adjusting the rank of experts across different layers could lead to improved performance. Consequently,

adaptively assigning the appropriate rank to these experts is a potential key focus of future research. Furthermore, while our method seeks to establish a unified multi-task model for practical applications, it is essential to consider potential out-of-distribution (OOD) issues that may arise during deployment, as they could impact the performance. To address this challenge and enhance the applicability of our work in real-world scenarios, future research could explore the integration of domain adaptation techniques [101, 102]. Investigating datasets such as WILDS [103], which are tailored for OOD challenges, could provide valuable insights and further improve performance across diverse contexts.

