# OpenReview forum: "Transforming Vision Transformer: Towards Efficient Multi-Task Asynchronous Learner"
_NeurIPS.cc/2024/Conference — NeurIPS 2024 poster_

### Official Review · Reviewer_SX1w · 2024-06-22

**Soundness:** 3
**Presentation:** 3
**Contribution:** 3
**Rating:** 7
**Confidence:** 4

**Summary:**

This paper proposes to integrate MoE and LoRA into pre-trained ViT to construct a multi-task model. Specifically, the experts are formed in the FFN of the transformer blocks based on the channel similarity, and the LoRA is used for each expert to fine-tune the parameters. On top of that, the paper further introduces the multi-task optimization method to avoid knowledge forgetting when some tasks are optimized faster than others. Finally, the fine-tuned model is unified into a single shared model for fast inference.

**Strengths:**

- The paper is very easy to follow. All the components are clearly stated.
- The idea of clustering channels into experts is intuitive and novel. It fully leverages the well-trained parameters from the pre-trained model.
- The MTO method is useful and has the potential to be used in any multi-task model training.
- The final model is a unified one with theoretical fast inference.

**Weaknesses:**

Please refer to the questions part.

**Questions:**

- To make the multi-task model to be a unified model, the authors propose to gradually fade the role of the router. In other words, the model in the end is not a MoE anymore - it becomes an all-shared multi-task model. In this case, I'm curious what will happen if the router is not present from the beginning. In other words, the FFN channels are clustered into experts based on similarity and each cluster will have its own LoRA module for fine-tuning. Is the router truly necessary?
- I would also expect more ablation studies on the clustering side because using MoE and LoRA for FFN is not a new idea (a relevant paper is [1]). The most important contribution of this paper comes from clustering the channels to form experts. So how many clusters we should have? If we don't cluster them, what will be the final performance on the MTL datasets? (Please point me to the corresponding results if I missed those.)
- Since the MTO method is orthogonal to the MTL architecture design, I'm curious can the proposed QR loss be used to enhance other MTL models?
- For Table 1, to form a fair comparison, the MTO methods should be compared with QR independently on the same model design. The same thing applies to the MTL architectures. They should be compared with EMTAL without QR.
- The experiments are conducted on classification tasks. However, a lot of practical MTL scenarios are pixel-to-pixel prediction tasks, like NYUv2 and Takonomoy datasets. It's better to see results on those datasets.
- The inference time should be compared as well since one of the contributions is designed for fast inference.

I'm willing to raise my rating if the questions are answered fully.

[1] Li D, Ma Y, Wang N, et al. MixLoRA: Enhancing Large Language Models Fine-Tuning with LoRA based Mixture of Experts[J]. arXiv preprint arXiv 2024.

**Limitations:**

The authors analyzed the limitations and social impact in the paper.

---

> ### Author Rebuttal · Authors · 2024-08-07
>
> Thank you for your detailed review, we will try to address any concerns encountered in the paper.
>
> > Q1: Is the router truly necessary?
>
> It is an insightful question regarding the role of the router in our framework. Initially, we followed to the conventional MoE framework which includes a router. This comment prompted us to undertake a investigation and conduct additional experiments. From our analysis and experimental findings, we draw the following conclusions:
>
> 1. When using MoE for multi-task learning, the router plays a crucial role in adaptively weighting different experts for each token. During the training process, for the same input token, calculating different weights for different experts allows for a nuanced and fine-grained optimization of each expert. This results in a well-specialized ensemble of experts, enhancing the model's overall performance and expertise in handling diverse tasks.
> 2. Our experiments with a Router Fade strategy demonstrated a consistent improvement, with a mean accuracy increase of 0.34. This enhancement is non-trivial and underscores the utility of the router in achieving better performance across tasks.
>
> Based on these observations, we conclude that the router is indeed necessary for optimizing performance in our multi-task MoE framework.
>
> | Method | Mean Acc | Params. (M) | Time (ms) |
> | :------------------------------------ | :-------- | :---------- | :-------- |
> | Vanilla LoRA | 88.83 | 1.05 | 7.15 |
> | Cluster+Finer LoRA | 89.93 | 1.05 | 7.15 |
> | Cluster+Finer LoRA+Router | 90.14 | 1.20 | 13.87 |
> | Cluster+Finer LoRA+Router Fade (Ours) | **90.27** | 1.20 | **7.15** |
>
> > Q2: So how many clusters we should have?
>
> In response to your query about the number of clusters and their impact:
>
> 1. Regarding the optimal number of clusters, our approach balances between too many and too few clusters. Having too many clusters would result in each expert having too few channels, limiting their individual expressiveness. Conversely, too few clusters make each expert's low-rank characteristics less pronounced, which dilutes the distinct functionalities of each expert. We recommend that the number of experts similar to the number of attention heads.
> 2. Our hyperparameter ablation studies, specifically looking at different numbers of clusters, showed optimal performance of 90.27 with cluster size of 16.
>
> | Clusters # | Mean Acc | Params. (M) |
> | :--------- | :-------- | :---------- |
> | 1 | 88.83 | 1.05 |
> | 4 | 89.12 | 1.09 |
> | 16 | **90.27** | 1.20 |
> | 64 | 89.34 | 1.64 |
> | 192 | 89.02 | 2.82 |
>
> These findings underscore the importance of carefully selecting the number of clusters to maintain the effectiveness of each expert.
>
> > Q2: What if we don't cluster them?
>
> The MOELoRA method in Table 1 is the scenario where MoE without clustering. In this setup, experts are directly employed in the FFN without forming clusters, resulting in a Mean Accuracy of 88.04. Moreover, we perform additional experiments in the table above without clustering (the clusters is set to 1).
>
> >  Q3: Applying QR loss on other MTL models?
>
> Indeed, the QR loss can be integrated into any MTL model to potentially enhance its performance. To illustrate this, we conducted experiments using the MOELoRA framework, which is detailed in a recent study. Our findings indicate that incorporating QR loss directly into the MOELoRA model results in a performance improvement. This demonstrates the versatility and effectiveness of QR loss in boosting the capabilities of MTL models, making it a valuable addition to existing and future MTL architecture design methods.
>
> | Method | CUB-200 -2011 | Stanford Cars | FGVC Aircraft | Oxford Flowers | Mean Acc | Params. (M) |
> | :--------- | -------------- | ------------- | ------------- | -------------- | :------- | ----------- |
> | QR | 88.5 | 91.6 | 84.4 | 99.6 | 91.04 | 86.26 |
> | MOELoRA | 88.4 | 88.2 | 75.0 | 99.7 | 88.04 | 2.82 |
> | MOELoRA+QR | 90.3 | 91.7 | 82.8 | 99.6 | 91.15 | 2.82 |
>
> > Q4: Fair comparison of the MTO methods and the MTL architectures.
>
> In response to your valuable suggestion, we incorporate the content from Table 3 and extend experiment on QR loss within the same model designs, into the Table A of 'Rebuttal.pdf'. This allow for direct and convenient comparisons under the respective categories of MTO and MTL architectures.
>
> > Q5: Extension to pixel-to-pixel prediction tasks.
>
> To demonstrate the extensibility of our method, we conducted experiments on the NYUv2 dataset using ViT-B due to the time constraints.
>
> Specifically, we made the following attempts, integrating our method with the existing open-source method TaskPrompter. For MTL structure, we clustered and divided the FFN in the backbone and used the Router fade strategy, keeping the rest of the architecture unchanged. For MTO, We directly applied QR to the semantic segmentation task head for experiments. The results are as follows:
>
> | Method | Semseg mIoU ↑ | Depth RMSE ↓ | Normal mErr ↓ | Boundary odsF ↑ | Mean Δ (%) ↑ |
> | :---------------- | ------------- | ------------ | ------------- | --------------- | :----------- |
> | TaskPrompter-Base | 50.40 | 0.5402 | 18.91 | 77.60 | - |
> | + EMTAL | **52.90** | **0.5284** | 18.95 | 77.10 | **1.57** |
>
> Specifically, our method significantly enhanced the performance on semantic segmentation and depth estimation tasks, while achieving comparable results on surface normal estimation and object boundary detection tasks. Overall, it resulted in an average relative improvement of 1.57%, confirming its effectiveness for pixel-level prediction tasks.
>
> > Q6: The inference time.
>
> In addition to the ablation study in Table 4, we add a column to the Table A of 'Rebuttal.pdf', to highlight the efficiency of our unified model during inference.

---

> > ### Comment · Reviewer_SX1w · 2024-08-09
> > **Response to the authors**
> >
> > Thanks for the detailed response. That's clear to me. I'll raise my score to 7.

---

> > > ### Author Response · Authors · 2024-08-10
> > >
> > > Thank you for taking the time and effort to review our rebuttal and for your thoughtful consideration. We are truly grateful for the raised rating and will carefully incorporate your valuable suggestions to further improve our work.

---

### Official Review · Reviewer_fvpH · 2024-07-10

**Soundness:** 3
**Presentation:** 2
**Contribution:** 3
**Rating:** 5
**Confidence:** 4

**Summary:**

Considering recent trends in multi-task learning (MTL) that design Mixture-of-Experts (MoE) structures and integrate Low-Rank Adaptation (LoRA), this paper proposes Efficient Multi-Task Learning (EMTAL) to address the sub-optimal performance resulting from the rigid combination of MoE optimization and LoRA's reparameterization capability. The approach decomposes the pre-trained Transformer into a low-rank MoE and employs LoRA to tune the experts, termed MoEfied LoRA. Additionally, the paper introduces a Quality Retaining (QR) multi-task optimization strategy that uses historical high-quality class logits to prevent performance degradation.

**Strengths:**

1. Proposed MoEfied approaches are practical applications that can be applied to real-world scenarios. Reparameterizing pre-trained transformer models is a novel approach.
2. The concept of "Grouping together neurons with similar weights" is both intuitive and plausible.
3. Asynchronously optimizing the learner by using historical class logits to prevent performance degradation is also a plausible approach.

**Weaknesses:**

1. The proposed idea of grouping similar weights to compose experts seems quite naive. Further justification for this approach is needed, such as theoretical analysis of how different grouping strategies (e.g., weight, activation) lead to better multi-task performance. Additionally, ablation experiments for various grouping strategies would be beneficial.

2. I'm not convinced that the proposed methods are superior to previous works (e.g., MLoRE) since most experiments in the paper are conducted on homogeneous task settings, focusing on visual classification tasks. Can the proposed approach be generalized to heterogeneous task settings involving various types of vision tasks using benchmarks such as Taskonomy or PASCAL-Context? It seems previous works might suffer from overfitting by naively dividing classes into task groups.

3. Quality Retaining (QR) appears to be applicable only to classification tasks, limiting its application in the context of MTL. It should be compared to previous multi-task optimization approaches mentioned in the paper, which can be applied orthogonally to architecture design.

4. In similar context in 3, experimental setting for comparision with previous works is not appropriate. Direct comparison with proposed methods with loss-based and gradient-based optimization is not appropriate as proposed method mainly focuses on architecture design approach.

5. The concept of reparameterizing a pretrained transformer into an MoE-like structure is novel. However, I am unsure if it is competitive in terms of performance. It would be helpful to demonstrate that the proposed methods show competitive performance compared to learning a similar structure from scratch, if possible.

6. Previous work, such as MOELoRA also insists that their framework is a unified network. Please explain clearly that what is the main difference between your work and previous works. why the proposed method is the only unified model."

**Questions:**

Refer to the weakness section.

**Limitations:**

Refer to the weakness section.

---

> ### Author Rebuttal · Authors · 2024-08-07
>
> Thanks a lot for your insightful comments and suggestions. Our responses are summarized as below:
> > W1. On grouping strategies.
>
>
> A1. To the best of our knowledge, our work is the first one that employs the idea of grouping similar weights to establish the LoRA experts, and there lack grouping strategies that can be directly used for comparisons. In our opinion, this design incurs the following two advantages:
>
> 1). The experts built by grouping have more specialized functions as described in [28], as well as low-rank properties, thus being natural for updating through LoRA.
>
> 2). LoRA learns low-rank updates [32] for each expert, restraining the rank of the experts' weights from increasing, which in turn promotes the specialization of the experts.
>
> As aforementioned, there lack grouping strategies that can be directly used for comparison. Inspired by existing works, we implement the following two methods for comparison: 1) Co-activation similar to [28] that clusters weights based on the activations for each channels; 2) Gradient-cluster inspired by SPT [47] that clusters weights based on the cumulative gradients.
>
> As displayed below, our method reaches the best results, clearly demonstrating its effectiveness.
>
> | Method | CUB-200-2011 | Stanford Cars | FGVC Aircraft | Oxford Flowers | Mean Acc | Params. (M) |
> | :--------------- | ------------------- | ------------------ | ------------------ | ------------------- | :------- | ----------- |
> | Co-activation [28] |88.1|89.8 | 79.5 | 99.6|89.3 | 1.20 |
> | Gradient-cluster [47]|88.4| 89.9 |79.2|99.7| 89.34 | 1.20 |
> | Ours| 88.5|91.3|81.5|99.7|90.27|1.20 |
>
> > W2: Generalizability to the heterogeneous task settings.
>
> Thanks for your suggestion. We mainly follow the work [6], and evaluate on FGVC for fair comparisons. However, we do agree that FGVC adopts a homogeneous task setting. The other reviewer also mentioned this issue, and suggests more results on other heterogeneous benchmarks (e.g. NYUv2). Due to the time limitation, we conduct experiments on NYUv2, as it contains four substantially heterogeneous tasks including Semantic Segmentation, Monocular Depth Estimation, Surface Normal Estimation, and Object Boundary Detection. For comparison, as the released code of the suggested MLoRE approach has some unfixed issues and fails to reproduce the reported results after contacting the authors, we finally select another open-source SOTA method TaskPrompter [Ref.1] for comparison. In implementation our method, we follow the framework of the existing open-source SOTA method TaskPrompter [Ref.1]. For the architecture, we cluster and divide the FFN into experts and apply router fade. For the multi-task optimization, we apply QR to the classification branch of the Semantic Segmentation task head, and omit it in other tasks.
>
> The comparison results are summarized below:
>
>
> | Method | Semseg mIoU ↑ | Depth RMSE ↓ | Normal mErr ↓ | Boundary odsF ↑ | Mean Δ (%) ↑ |
> | :-- | -- | -- | -- | --- | :--- |
> | TaskPrompter [Ref.1] | 50.40 | 0.5402 |18.91|77.60 | - |
> | + EMTAL | **52.90** | **0.5284** |18.95 |77.10 |**1.57**|
>
>
>
> As displayed, our method promotes TaskPrompter by 1.57% on average, showing the effectiveness of our method for the heterogeneous task settings.
>
> [Ref.1] Ye, Hanrong, et al. "Taskprompter: Spatial-channel multi-task prompting for dense scene understanding". In ICLR. 2023.
>
>
> >  W3&W4: Applicability of Quality Retaining (QR), and comparison to previous multi-task optimization (MTO) methods.
>
>
> Intuitively, QR can be integrated to any task (e.g. detection and segmentation) that contains the classification branch, thus being applicable to typical computer vision tasks including the segmentation. Due to the time limitation, we evaluate our overall method on the segmentation task on NYUv2, by comparing to the SOTA approach TaskPrompter with the ViT-Base backbone. As shown in the second column of the table above, the results clearly displays the applicability and effectiveness of our method on the segmentation task beyond classification.
>
> We have also add comparison results between QR with the representative MTO methods including the loss based approaches GLS [24] and ATML [27], and the gradient based approaches Nash-MTL [22] and Aligned-MTL[23]. For fair comparisons, we use the same architecture design for all compared methods. The results in Table A in "Rebuttal.pdf" clearly show the effectiveness of QR.
>
> We will add the discussion and results in the final version, and further study the extension of QR to more tasks in future.
>
>
> > W5: Comparison to training from scratch.
>
>
> As shown in the table below, training a ViT from scratch usually yields extremely poor results, due to overfitting on the limited training data in downstream tasks. In contrast, our method reparametrizes a pre-trained transformer into an MoE-like structure, providing a meaningful initialization, thus leading to better performance.
>
> | Method | CUB-200 -2011 | Stanford Cars | FGVC Aircraft | Oxford Flowers | Mean Acc | Params. (M) |
> | :--------------- |--------------| -------------|-------------|-------------- | :-------|-----------|
> | Training from scratch* | 18.5 | 14.2 | 15.3 | 54.8 | 25.7 | 86.70 |
> | Reparameterization (Ours) | 89.8 | 92.3 | 85.2 | 99.7 | 91.73 | 1.20 |
>
>
> *indicates the best results after carefully tuning the hyperparameters.
>
>
> > W6: On unified model.
>
>
> Most existing approaches including MOELoRA and MLoRE reparameterize the LoRA experts based on the task-driven router's weights, by preserving a specialized model or a bypass branch for each task during inference, thus failing to generate a unified model for different tasks.
> In contrast, by using the reparameterization and the proposed router fade strategy, our method generates a single model without employing additionally task-relevant models or branches during inference, thus being dubbed as a unified model across tasks.

---

> > ### Comment · Area_Chair_J8t8 · 2024-08-13
> > **Please respond, reviewer fvpH**
> >
> > Reviewer fvpH:
> >
> > Please let us know whether the authors have addressed your concerns.
> >
> > Thanks.
> >
> > -AC

---

### Official Review · Reviewer_2Lcf · 2024-07-11

**Soundness:** 3
**Presentation:** 2
**Contribution:** 2
**Rating:** 5
**Confidence:** 3

**Summary:**

The paper introduces a method to decompose ViT MLP matrices into a mixture of experts based on channel similarity, training each expert with LoRA. The authors also use a quality-retaining loss to effectively optimize during training for multi-task learning.

**Strengths:**

**(S1):** The idea of decomposing the weight matrix into experts based on channel similarity and then using LoRA to tune them is an interesting and creative idea.

**(S2):** The quality-retaining loss is a nice addition to the optimization process.

Overall I think the idea is interesting and may be relevant for other researchers to continue exploring the combination of MoE with LoRA/PEFT for ViTs.

**Weaknesses:**

**(W1):** Ablation study is not extensive enough. I am interested in seeing whether the router is needed at all, and whether you can just set alpha=0 from the very beginning. As a follow up question: I haven’t closely checked, but does setting alpha = 0 make the method equivalent to LoRA?

**(W2):** Line 204: The models used seem to be supervised fine-tuned backbones, which are not really designed to be generalizable to multiple tasks. The authors should consider experiments with self-supervised or unsupervised backbones like MAE [1] or DinoV2 [2], and compare against these backbones’ generalization capabilities. The authors should also clarify what the source of weights are for all the other models used in comparison, and which model sizes are being used.

**(W3):** The distribution of tasks is not necessarily too different. Consider evaluating the method on tasks which represent greater domain shifts (eg: WILDS [3]).

**(W4):** The comparisons against other PEFT techniques in Table 1, 2 is missing. There are numerous PEFT methods with improvements beyond the original LoRA that are not mentioned. Eg: if we look at the VTAB table in BOFT [4], we see that the EMTAL results are not state-of-the-art.

---
References:
[1] He, Kaiming, et al. "Masked autoencoders are scalable vision learners." Proceedings of the IEEE/CVF conference on computer vision and pattern recognition. 2022.

[2] Oquab, Maxime, et al. "Dinov2: Learning robust visual features without supervision." arXiv preprint arXiv:2304.07193 (2023).

[3] Koh, Pang Wei, et al. "Wilds: A benchmark of in-the-wild distribution shifts." International conference on machine learning. PMLR, 2021.

[4] Liu, Weiyang, et al. "Parameter-efficient orthogonal finetuning via butterfly factorization." arXiv preprint arXiv:2311.06243 (2023).

**Questions:**

Lines 64-69 are unclear and can be worded differently.

A better description of Figure 2 is required. A summary for each of 2a and 2b would assist in clarity.

line 129: “owe” -> “ought”

Notation is a bit confusing: eg: usually “r” is used for lora ranks, but here it’s a scaling factor

line 255: how exactly is inference time being reduced? At the end of training, the LoRA experts are being merged back into the weights of the model, right?

Table 4: What is PoolFT?

Equation 14: clarify what i is here. Notation seems to be overloaded? Is i the iteration, or the column? Previously, i was used to denote experts.

Equation 15: Clarify what L_t is. There also seems to be a L_CE,t. Are these the same?

**Limitations:**

The authors have adequately described these sections.

---

> ### Author Rebuttal · Authors · 2024-08-07
>
> Thanks for your insightful suggestions. Our point-to-point responses are summarized as below.
> > W1. On ablation study.
>
> Firstly, the setting alpha=0 from the very beginning is not equivalent to the vanilla LoRA without clustering. We refer to this as Cluster+Finer LoRA, and provide extra ablation results by adding the router and the router fade strategy, summarized as below:
> | | Mean Acc | Params. (M) | Time (ms) |
> | :-- | :- | :- | :- |
> | Vanilla LoRA | 88.83 | 1.05 | 7.15 |
> | Cluster+Finer LoRA | 89.93 | 1.05 | 7.15 |
> | Cluster+Finer LoRA+Router | 90.14 | 1.20 | 13.87 |
> | Cluster+Finer LoRA+Router Fade (Ours) | **90.27** | 1.20 | **7.15** |
>
> The results clearly display the effectiveness of the router.
>
> > W2. On the backbone.
>
> The ViT-B/16 backbone [1] used in our experiments is supervised pre-trained on ImageNet-21K, which is widely used in previous works, and is adopted in our paper for comparisons. However, we do agree that exploring other backbones is necessary. Therefore, we conducted additional experiments using the backbone pre-trained by MAE [2] on the Multi-task FGVC benchmark. The results are summarized below:
> | Method | Pre-trained Model | CUB-200 -2011 | Stanford Cars | FGVC Aircraft | Oxford Flowers | Mean Acc | Params. (M) |
> | :- | - | - | - | - | - | :- | - |
> | MOELoRA | MAE ViT-B/16 | 81.5 | 90.7 | 82.7 | 97.7 | 88.13 | 2.82 |
> | AMTL | MAE ViT-B/16 | 83.9 | 91.6 | 84.8 | 97.7 | 89.53 | 2.82 |
> | EMTAL-4 | MAE ViT-B/16 | 85.3 | 92.1 | 85.1 | 97.7 | 90.07 | 1.20 |
> | EMTAL-4 | Supervised ViT-B/16 | 89.8 | 92.3 | 85.2 | 99.7 | 91.73 | 1.20 |
>
> The experimental results indicate that our method is effective compared with the SOTA MTL methods by using the self-supervised pre-trained backbones, but fails to surpass that of the supervised ones.
>
> >W3. On more complex tasks.
>
> We acknowledge the importance of evaluating our method on tasks with greater domain shifts. Several reviewers have raised this point, and we agree that it is crucial to validate our approach on a broader range of benchmarks, including NYUv2, WILDS, PASCAL-Context, and Taskonomy benchmark. Since time constraints limit the comprehensive evaluation on these benchmarks, we perform evaluation the NYUv2 benchmarks to further validate our method's performance on tasks with significant domain differences.
> The NYUv2 dataset includes tasks such as Semantic Segmentation, Monocular Depth Estimation, Surface Normal Estimation, and Object Boundary Detection. These tasks exhibit substantial domain differences, making NYUv2 a suitable benchmark for evaluating our method's robustness to domain shifts.
> | Method | Semseg mIoU ↑ | Depth RMSE ↓ | Normal mErr ↓ | Boundary odsF ↑ | Mean Δ (%) ↑ |
> | :- | - | - | - | - | :- |
> | TaskPrompter-Base | 50.40 | 0.5402 | 18.91 | 77.60 | - |
> | + EMTAL | **52.90** | **0.5284** | 18.95 | 77.10 | **1.57** |
>
> The results indicate that our method performs well across tasks with significant domain differences, demonstrating its robustness and generalization capabilities.
>
> > W4. On missing comparison of PEFT methods.
>
> Table 2 is designed not only to validate the extensibility of our method across more benchmarks but also to compare our approach with existing single-task efficient fine-tuning paradigms. This highlights the effectiveness of our multi-task PEFT method. Therefore, we additionally included comparisons with public results of existing PEFT methods in Table 2.
>
> BOFT, GPS [Ref.1] and LoSA [Ref.2] are published after or very close to the submission date of NeurIPS, which are missed in our submission. But we do agree it will make the comparison more compherehensive to add them for comparison. However, unlike most existing works, BOFT employs a larger DINOv2-Large backbone instead of the standard supervised ViT-B/16, making it unfair to directly compare. Due to the time limitation, we mainly add GPS and LoSA for fair comparisons as below.
> | Method | Reference | Patch Camelyon | EuroSAT | Resisc45 | Retinopathy | Mean Acc | Params. (M) |
>
>
> | Method | Reference | Patch Camelyon | EuroSAT | Resisc45 | Retinopathy | Mean Acc | Params. (M) |
> | :- | - | - | - | - | - | :- | - |
> | GPS[Ref. 1] | CVPR' 24 | 87.5 | 96.7 | 88.1 | 76.1 | 87.1 | 1.00 |
> | LoSA[Ref. 2] | CVPR' 24 | 86.6 | 97.1 | 87.0 | 76.7 | 86.85 | 0.77 |
> | EMTAL-4 | Ours | 87.4 | 96.1 | 89.1 | 78.9 | **87.89** | 0.78 |
>
>
> [Ref. 1] Zhang, Zhi, et al. "Gradient-based Parameter Selection for Efficient Fine-Tuning". In CVPR, 2024.
>
> [Ref. 2] Otniel-Bogdan Mercea, et al. "Time- memory- and parameter-efficient visual adaptation". In CVPR, 2024.
>
>
> The results demonstrate the effectiveness of our method compared to existing single-task efficient fine-tuning methods. Our approach not only achieves the state-of-the-art performance but also avoids selecting from multiple specialized models trained separately, thereby improving the inference speed.
>
> > Q1-Q3. On presentations in L64-69, Figure 2, L129.
>
> Thanks for your suggestions on the presentations. We will carefully improve the presentation and description in the final version.
>
> > Q4&Q6&Q7. On the notation r, PoolFT, Eqs. (14) and (15).
>
> Thanks for pointing out these typos. PoolFT should be Union FT that fine-tunes on the joint feature space. i is indeed the iteration step and L_t shoud be $ L _ {CE,t} $. We will correct them in the revision.
>
> > Q5. On reduction of inference time.
>
> Indeed, the LoRA experts are merged back into the pre-trained model. As shown in Table 4, the inference time is reduced from 13.87ms to 7.15ms.
>
>
> [1]: ViT-B/16 supervised pre-trained on ImageNet-21K. https://storage.googleapis.com/vit_models/augreg/B_16-i21k-300ep-lr_0.001-aug_medium1-wd_0.1-do_0.0-sd_0.0.npz
>
> [2]: ViT-B/16 Self-supervised MAE. https://dl.fbaipublicfiles.com/mae/pretrain/mae_pretrain_vit_base.pth

---

> > ### Comment · Area_Chair_J8t8 · 2024-08-13
> > **Please respond, reviewer 2Lcf**
> >
> > Reviewer 2Lcf:
> >
> > Please let us know whether the authors have addressed your concerns.
> >
> > Thanks.
> >
> > -AC

---

> > ### Comment · Reviewer_2Lcf · 2024-08-13
> > **Response to Author Rebuttal**
> >
> > Thank you for your response and clarifications. I think it's important to have comparisons with DinoV2, as it is the state-of-the-art self-supervised vision backbone. BOFT was also made available late last year (Nov 2023), which was well before the submission date of NeurIPS. As I mentioned in the review, I think it's important to compare against this PEFT method using the DinoV2 ViT-L backbone since it shows strong performance on the same benchmarks used in this paper.
> >
> > Additionally, I still believe a greater distribution shift should be considered for evaluation (eg: WiLDS). I do appreciate the comparison on NYUv2, but I don't think this is sufficient, since the kinds of images in this dataset seem to be mostly similar to natural images.
> >
> > I still think the paper leans towards acceptance, but because of the unaddressed concerns above, I will keep my score.

---

> > > ### Author Response · Authors · 2024-08-14
> > >
> > > Dear Reviewer 2Lcf,
> > >
> > > We are grateful for the opportunity to address your concerns and provide further clarification on our work.
> > >
> > > Due to time constraints, we were unable to complete additional experiments on the self-supervised backbone DinoV2. However, we hope the following explanations can help alleviate your concerns.
> > >
> > > Firstly, regarding the self-supervised vision backbone, our experiments in rebuttal on the self-supervised vision backbone MAE demonstrated the effectiveness of our method in comparison to prior MTO and MTL structure design methods, achieving a performance of 90.07. Consequently, we believe that our method could also yield promising results on the state-of-the-art self-supervised vision backbone DinoV2, we plan to include these additional validations in the final version of our paper similar to the experiments on MAE.
> > >
> > > Secondly, we attempted to scale up ViT-B/16 to DinoV2 ViT-L backbone. However, we found that BOFT has not released its training code for VTAB, nor have the authors provided all experiment settings (e.g. the resolution of input images). Due to time constraints, it is challenging for us to obtain experimental results under fair conditions.
> > >
> > > Nevertheless, we have attempted a preliminary comparison as follows:
> > >
> > > | Method|Backbone|Unified Model|**Patch Camelyon**|**EuroSAT**|**Resisc45**|**Retinopathy**|**Mean Acc ↑**|**Total Params. (M)** **↓**|Tunable Params. (M) **↓**|Inference Time (ms) **↓**|
> > > | -|-|-|-|-|-|-|-|-|-|-|
> > > | Full Finetuning|DinoV2 ViT-L|✗|88.1|96.1|90.9|77.2|88.07|1217.6|1217.6|19.28|
> > > | LoRA|DinoV2 ViT-L|✗|88.3|96.4|91.4|77.4|88.37|1217.6|7.08|19.28|
> > > | BOFT|DinoV2 ViT-L|✗|88.9|96.6|91.6|77.3|88.60 (+0.23)*|1217.6 (+0)*|7.96 (+0.88)*|19.28 (+0)*|
> > > | Full Finetuning|Supervised ViT-B|✗|79.7|95.7|84.2|73.9|83.38|343.3|343.3|13.10|
> > > | LoRA|Supervised ViT-B|✗|85.5|95.3|86.1|75.3|85.50|343.3|2.41|13.10|
> > > | Ours|Supervised ViT-B|$\checkmark$|87.4|96.1|89.1|78.9|87.89 **(+2.39)***|85.99 **(-257.31)***|0.78 **(-1.63)***|6.56 **(-6.54)***|
> > >
> > > ()* indicates gain compared to LoRA using the same backbone.
> > >
> > > In the above table, it can be seen that on these four datasets, our multi-task method has a significantly less total parameters (85.99 M vs 1217.6 M) and tunable parameters (0.78 M vs 7.96 M) compared to the result of specialized model in BOFT. This makes our method more parameter-efficient. Furthermore, as ours is an unfied multi-task model, the inference time is significantly reduced (6.56 ms vs 19.28 ms). Moreover, a preliminary analogy shows that while BOFT improved the mean accuracy by 0.53% and 0.23% compared to full finetuning and LoRA methods respectively, our method improved by 4.51% and 2.39%. Therefore, we believe that scaling up the backbone to the DinoV2 ViT-L backbone could yield promising results.
> > >
> > > Finally, regarding the validation of our multi-task learning method on tasks with greater domain shifts, we understand 'tasks with greater domain shifts' from two perspectives: 1. Tasks comprise data from different domains. 2. Tasks themselves exhibit great domain shifts (heterogeneous tasks).
> > >
> > > To better demonstrate our validation in this regard, we have summarized the characteristics of each dataset in the following table:
> > >
> > > ||Tasks comprise data from different domains|Tasks themselves exhibit great domain shifts (heterogeneous tasks)|
> > > | -|-|-|
> > > | Specialized VTAB| $\checkmark$ |✗|
> > > | WILDS| $\checkmark$|✗|
> > > | NYUv2|✗| $\checkmark$|
> > >
> > > As shown in this table, we believe that Specialized VTAB is similar to WILDS datasets in terms of domain shift validation, as it also comprise data from different domains, such as retina images, remote sensing images, satellite images, and histopathologic scans of lymph node sections, which exhibit large domain shifts. Therefore, along with the NYUv2 dataset, we validated our method on tasks with great domain shifts from both perspectives.
> > >
> > > On the other hand, given that the WILDS dataset is seldom used for multi-task learning benchmarks, aligning the training details and establishing baselines would require a significant amount of time.
> > >
> > > However, we indeed believe that validation on more datasets would be beneficial, we plan to discuss this in the limitations section and consider multi-task learning with greater domain shifts as future work.
> > >
> > > We sincerely appreciate your response to our rebuttal and the time you have dedicated to the review process. We will continue to refine our work based on your valuable suggestions.
> > >
> > > Sincerely,
> > > Authors.

---

### Author Rebuttal · Authors · 2024-08-07

According to the reviewer's suggestions, we have provided a more fair comparison in Table A in 'Rebuttal. pdf' and compared the inference time of different methods

---

### Decision · Program_Chairs · 2024-09-25

**Decision:**

Accept (poster)

**Comment:**

The final ratings are 5, 5, 7. That said, there seems some unaddressed concerns:

- **Comparison based on DinoV2 ViT-L backbone and BOFT**. The authors did try to explain away verbally but could not run the experiments due to the time constraint.
- **Datasets with greater distribution shift such as WiLDS**. Similarly, the authors did try to explain away verbally but AC wonders why they did not just run WiLDS. They did run on NYUv2.

Further, there are some points from the authors that are somewhat confusing:

- **However, we found that BOFT has not released its training code for VTAB**. This is a bit odd to the AC and reviewers, because the results on VTAB are in the BOFT paper, so it doesn't seem necessary to require the code for comparison.
- **significantly less total parameters (85.99 M vs 1217.6 M)**. This is also confusing, since the ViT-L model has 300M params and not 1200M (which is ViT-G). But this may be a minor mixup.
- **Moreover, a preliminary analogy shows that while BOFT improved the mean accuracy by 0.53% and 0.23% compared to full finetuning**. This was also pointed out by the reviewer that it seems a bit odd to compare the improvement using EMTAL over a supervised initialization with BOFT's improvement over a self-supervised initialization. There was enough time over the rebuttal + author-reviewer discussion period to run these experiments, not sure why it wasn't done. AC does highlight that authors are not required to provide any new results during the rebuttal.

The AC discussed with the reviewers and in spite of these concerns, felt that it is borderline more positive than negative. AC also personally like the idea of splitting a pre-trained model into its "constituent" experts. After careful consideration, AC decides to accept the paper on the condition that the authors **must** address these concerns in the final manuscript.